# Association between different metabolic obesity phenotypes and colorectal adenoma

Li Lin[1], Yuhan Ying[1], Long Shu[2], Xiaoling Lv[3], Qin Zhu[2,4]*

1 The Second School of Clinical Medicine, Zhejiang Chinese Medical University, Hangzhou, Zhejiang, China, 2 Department of Clinical Nutrition, Zhejiang Hospital, Hangzhou, Zhejiang, China, 3 Zhejiang Provincial Key Lab of Geriatrics and Geriatrics Institute of Zhejiang Province, Zhejiang Hospital, Hangzhou, Zhejiang, China, 4 Department of Gastroenterology, Zhejiang Hospital, Hangzhou, Zhejiang, China

* zhuqin-1@163.com

## Abstract

### Background

Obesity and metabolic abnormalities are independently and interactively associated with colorectal adenoma (CRA). This study aimed to investigate the association between different metabolic obesity phenotypes and CRA, and to assess whether this relationship is influenced by stratification based on age and sex.

### Patients and methods

A total of 2042 subjects were enrolled in this study. The patients were classified into four metabolic obesity phenotypes based on their body mass index (BMI) and metabolic status, including metabolically healthy non-obesity (MHNO), metabolically unhealthy non-obesity (MUNO), metabolically healthy obesity (MHO), and metabolically unhealthy obesity (MUO). Multiple logistic regression analysis and further subgroup analysis based on sex and age were performed to evaluate the association between the metabolic obesity phenotypes and the occurrence of CRA.

### Results

In the overall population, CRA was detected at significantly higher rates in both the MUNO and MUO phenotypes compared to the MHNO and MHO phenotypes. After adjusting for confounders, in the overall population, the MUNO (OR = 1.353, 95% CI: 1.099–1.666, P < 0.05) and MUO (OR = 1.558, 95% CI: 1.111–2.187, P < 0.05) phenotypes were identified as risk factors for the development of CRA compared to the MHNO phenotype. Subgroup analysis based on sex revealed that in female subjects, the MUNO phenotype (OR 1.694; 95% CI 1.243–3.308, p < 0.01) exhibited significantly elevated risks of CRA compared to those in the MHNO phenotype, whereas the MHO phenotype showed no significant differences. Furthermore, in the male subjects, no statistically significant differences were observed among the

**Data availability statement:** All relevant data are within the manuscript and its Supporting information files.

**Funding:** This work was supported by the National Natural Science Foundation of China (No.82004040).Qin Zhu(No.82004040) participated in methodology and supervision of this work.

**Competing interests:** The authors have declared that no competing interests exist.

four groups. Subgroup analysis based on age revealed that under 60 years with the MUNO (OR = 1.648, 95% CI: 1.252–2.169, P < 0.01) phenotype and over 60 years with MUO (OR = 2.301, 95% CI: 1.252–4.348, P < 0.01) phenotype, had significantly increased risks of CRA.

## Conclusion

Metabolically unhealthy phenotypes are associated with a higher risk of CRA incidence. Clinical screening for CRA should focus on metabolically unhealthy subjects.

---

## Introduction

Colorectal adenoma (CRA) is the most common benign tumor of the colorectum and has been recognized as a precancerous lesion of colorectal cancer (CRC) [1]. CRA accounts for about 70–80% of CRC cases, particularly in cases with advanced adenomas [2]. CRC is the second most common cause of cancer-related deaths worldwide and the third most common type of cancer overall [3]. Statistics have shown a growing incidence of CRC among younger age groups [4]. The annual rise in CRC cases in China has resulted in a significant financial burden for the country [5]. Additionally, due to the insidious onset of CRC, the disease is frequently diagnosed at an advanced stage. Among all early screening techniques, total colonoscopy remains the gold standard for CRC diagnosis [6], enabling prompt treatment of early lesions and direct access to pathologic samples. Following the advent of colonoscopic treatments, endoscopic CRA removal has significantly decreased the mortality rate of CRC [7]. Determining the risk factors for CRA and reliably identifying subjects at increased risk for CRA may improve the prevention and intervention strategies.

Studies have shown that age, sex, family history, weight, lifestyle, diet, and medications are known risk factors for CRA [8]. Obesity has emerged as a serious global public health concern, driven by changes in diet and routines and posing a significant risk for CRC [9]. A retrospective case-control study reported a significant association between overweight or obesity and advanced adenomas [10]. Obesity typically coexists with other metabolic problems, such as low levels of high-density lipoprotein cholesterol, excessive triglycerides, and insulinemia [11]. These metabolic diseases also affect the development of CRA via their association with obesity. Obese individuals may exhibit varying metabolic abnormalities. However, not all individuals with metabolic disorders are overweight or obese, and not all overweight or obese individuals suffer from metabolic diseases. Consequently, the concept of metabolically healthy obesity (MHO), which provides a comprehensive description of the body's metabolic processes and overall physical health, was originally put forth by Sims in 2001 [12]. A growing number of studies have shown that individuals classified as MHO may be exposed to a higher risk of cardiovascular diseases, prostate cancer, and erosive esophagitis compared to metabolically healthy non-obesity (MHNO) counterparts [13–15].

Previous studies have explored the association between various metabolic obesity phenotypes and gastrointestinal polys [16]. The complexity of the interaction between

obesity and metabolic disorders and its impact on patients with CRA remains poorly understood. However, the differential role of obesity and metabolic health phenotypes on CRA in the Chinese population has not yet been fully evaluated. This cross-sectional cohort study of a Chinese population aimed to elucidate the relationship between several metabolic obesity phenotypes and the incidence of CRA. The findings could be used to identify individuals at high risk for CRA, enhance the understanding of the role of obesity phenotypes in CRA development, and lay a foundation for CRC prevention and clinical intervention.

## Materials and methods

### Data source

Participants in this retrospective analysis were those who had received a complete medical examination at Zhejiang Hospital. The study recruited 6689 participants between1/1/2023 to 31/8/2024, who had comprehensive medical examinations, including physicals, blood tests, and colonoscopies.Authors completed data collection between 1/10/2024 and 10/2/2025 after obtaining ethical approval from the Ethics Committee of Zhejiang Hospital, had no access to information that could identify individual participants during or after data collection.Exclusion criteria include: (1) Age under 18 years; (2) HIV infection, pregnancy, severe organ failure, and active gastrointestinal cancer;(3) absence of information on body mass index (BMI), systolic blood pressure (SBP),diastolic blood pressure (DBP), fasting blood glucose (FBG), triglyceride(TG),total cholesterol (TC),low-density lipoprotein cholesterol(LDL-C) or high-density lipoprotein cholesterol (HDL-C) at admission; (4) have inflammatory bowel illness (such as Crohn's disease and ulcerative colitis) or hereditary CRC syndrome (such as familial adenomatous polyposis).This study was conducted according to the Declaration of Helsinki as revised in 2013, and the protocol was approved by the Ethics Committee of Zhejiang Hospital (Approval NO: 2024 Clinical Trial Approval No.114K). The need for patient consent was waived due to the retrospective nature of the study.

### Data collection and measurement

Eligible nurses followed predefined procedures to record demographic information such as age, gender, family history, smoking and alcohol usage, past medical history, and medications taken. All patients' blood pressure was recorded in the seated position using a digital sphygmomanometer after resting for at least ten minutes. The patients were fasted for at least ten hours, and blood samples were collected. The blood samples were analyzed at the Medical Laboratory Center of Zhejiang Hospital, including serum biochemical indices such as fasting blood glucose, total cholesterol, high-density lipoprotein, and triglycerides.

### Colonoscopy

All patients received a liquid diet for 24 hours before the examination, and polyethylene glycol was used to prepare their bowels as part of the usual protocol. In our hospital, a specialized endoscopist performed a colonoscopy, and all lesions were endoscopically removed. The biopsy tissues were viewed under a microscope by a skilled pathologist to categorize polyps according to pathologic characteristics.

### Definition of the metabolic obesity phenotypes

The metabolic profiles of the individuals were analyzed according to the Adult Treatment Panel III (ATP-III) criteria [17] and the China Guidelines for Type 2 Diabetes [18]. The parameters for determining metabolically unhealthy individuals were established based on the following criteria: (1) high blood pressure (SBP ≥ 130 mmHg or DBP ≥ 85 mmHg) or taking antihypertensive medical treatments; (2) hyperglycemia (FBG level ≥ 5.60 mmol/L) or taking antidiabetic medical treatments; (3) fasting TG ≥ 1.7 mmol/L or taking lipid-lowering medical treatments; (4) fasting HDL-C < 1.04 mmol/L or taking lipid-lowering medical treatments.

The BMIs of all the enrolled individuals were determined by dividing the weights of the individuals by the square of their heights ($kg/m^2$). Obesity was defined as BMI ≥ 28 $kg/m^2$ based on the criteria developed by the Working Group on Obesity in China [19].

The participants were classified into four groups based on these criteria: (1) MHO, people with BMI ≥ 28 $kg/m^2$ and one or no indicators of metabolically unhealthy; (2) MUO, people with BMI ≥ 28 $kg/m^2$ and two or more indicators of metabolically unhealthy; (3) MHNO, people with BMI < 28 $kg/m^2$ and one or no indicators of metabolically unhealthy; and (4) MUNO, people with BMI < 28 $kg/m^2$ and two or more indicators of metabolically unhealthy.

## Statistical analysis

All of the statistical analyses were carried out by SPSS (version 23.0) and the R software (version 4.4.2). The Shapiro-Wilk test was conducted to assess whether continuous variables conformed to a normal distribution. Continuous data were presented as means ± standard deviations (SD) or median (interquartile range), and comparisons were made using one-way analysis of variance for normally distributed data and the Whitney U-tests for skewed distributions. Categorical variables were presented as frequencies and percentages, and differences between groups were evaluated using the chi-squared or Fisher's exact tests as appropriate. Comparisons across more than two groups were performed using one-way analysis of variance (ANOVA), followed by Tukey's test for between-group comparisons. The odds ratio (OR) and 95% confidence interval (CI) of the risk of CRA in each metabolic obesity phenotype group were calculated by logistic regression analysis. Given the multiple comparisons performed across different subgroups (sex, age), we applied a more stringent significance level ($P < 0.01$) for interpreting the results of subgroup analyses to reduce the risk of Type I error. The primary analysis in the overall population was interpreted at $P < 0.05$.

## Results

### Comparative analysis of baseline data between the non-adenomas and adenomas groups

The behavioral and biochemical characteristics of all 2042 subjects, comprising 1049 males and 993 females, were collected and analyzed, as shown in Table 1. In the overall population, subjects diagnosed with CRA exhibited significantly higher levels of BMI, SBP, DBP, TG, and FBG compared to those without CRA. In addition, a higher prevalence of diabetes and hypertension, elevated rates of smoking and alcohol consumption, and a greater risk of developing CRA were observed (all $P < 0.05$). Furthermore, adenoma patients demonstrated reduced HDL levels, though no significant differences were observed in LDL-C or TC between the two groups.

The occurrence of adenomas was 54.2% and 30.9% among male and female subjects, respectively. Among female subjects, those with adenomas displayed elevated BMI, SBP, and TG levels, as well as lower HDL levels compared to non-adenoma counterparts (all $P < 0.05$). However, no significant differences in smoking or alcohol consumption behaviors were observed. In male subjects, significant differences were observed only in SBP and FBG levels, whereas other indicators showed no significant differences.

### Comparative analysis of baseline data between different metabolic obesity cohorts

This study explored the baseline characteristics of female participants (n = 993) across distinct metabolic obesity phenotypes, as shown in Table 2. The female individuals enrolled in this study had an average age of 57 years. The MHNO, MHO, MUNO, and MUO cohorts comprised 582 (58.6%), 27 (2.7%), 339 (34.1%), and 45 (4.5%) of the female subjects, respectively. Moreover, the frequency of adenomas in these cohorts was 23.5%, 33.3%, 41.6%, and 44.4%, respectively (p < 0.05; Fig 1A). Compared with participants with the MHNO or MHO phenotypes, those classified as MUNO and MUO phenotypes displayed a significantly elevated occurrence of CRA ($P < 0.05$). Furthermore, CRA incidence increased

**Table 1. Comparison of baseline characteristics of enrolled subjects with and without adenoma.**

| Variables | All subjects | | | Females | | | Males | | |
|---|---|---|---|---|---|---|---|---|---|
| | No Adenoma | Adenoma | P-value | No Adenoma | Adenoma | P-value | No Adenoma | Adenoma | P-value |
| No. of cases | 1166 | 876 | – | 686 | 307 | – | 480 | 569 | – |
| Age, years | 54.00 [46.00, 61.00] | 59.00 [53.00, 67.00] | <0.001 | 55.00 [47.25, 62.00] | 60.00 [54.00, 67.00] | <0.001 | 53.00 [45.00, 60.00] | 59.00 [53.00, 67.00] | <0.001 |
| Sex (male), n (%) | 480 (41.2) | 569 (65.0) | <0.001 | – | – | – | – | – | – |
| Smoking, n (%) | 189 (16.2) | 257 (29.3) | <0.001 | 8 (1.2) | 4 (1.3) | >0.050 | 181 (37.7) | 253 (44.5) | 0.032 |
| Drinking, n (%) | 188 (16.1) | 244 (27.9) | <0.001 | 28 (4.1) | 10 (3.3) | 0.655 | 160 (33.3) | 234 (41.1) | 0.011 |
| Diabetes, n (%) | 89 (7.6) | 126 (14.4) | <0.001 | 42 (6.1) | 34 (11.1) | 0.010 | 47 (9.8) | 92 (16.2) | 0.003 |
| Hypertention, n (%) | 301 (25.8) | 354 (40.4) | <0.001 | 144 (21.0) | 111 (36.2) | <0.001 | 157 (32.7) | 243 (42.7) | 0.001 |
| BMI, kg/m$^2$ | 23.44 [21.34, 25.69] | 24.22 [22.29, 26.23] | <0.001 | 22.48 [20.70, 24.64] | 23.62 [21.66, 25.07] | <0.001 | 24.64 [22.84, 26.96] | 24.57 [22.66, 26.57] | 0.316 |
| SBP, mmHg | 125.00 [115.00, 136.00] | 130.00 [119.00, 141.00] | <0.001 | 123.00 [112.00, 135.00] | 130.00 [119.00, 141.00] | <0.001 | 128.00 [118.75, 137.00] | 130.00 [118.00, 141.00] | 0.021 |
| DBP, mmHg | 79.00 [72.00, 86.00] | 80.00 [72.75, 87.00] | 0.011 | 76.00 [69.00, 83.75] | 78.00 [71.00, 84.00] | 0.056 | 82.50 [75.00, 90.00] | 81.00 [74.00, 89.00] | 0.149 |
| TC, mmol/L | 4.84 [4.23, 5.51] | 4.84 [4.18, 5.44] | 0.197 | 4.92 [4.29, 5.63] | 5.12 [4.49, 5.74] | 0.039 | 4.74 [4.15, 5.39] | 4.70 [4.07, 5.29] | 0.129 |
| TG, mmol/L | 1.31 [0.91, 1.97] | 1.58 [1.07, 2.34] | <0.001 | 1.15 [0.83, 1.65] | 1.51 [1.04, 1.93] | <0.001 | 1.62 [1.09, 2.49] | 1.66 [1.12, 2.57] | 0.770 |
| HDL-C, mmol/L | 1.23 [1.04, 1.48] | 1.17 [0.98, 1.39] | <0.001 | 1.35 [1.16, 1.61] | 1.31 [1.12, 1.52] | 0.010 | 1.09 [0.93, 1.26] | 1.10 [0.93, 1.28] | 0.582 |
| LDL-C, mmol/L | 2.77 [2.24, 3.30] | 2.72 [2.20, 3.22] | 0.167 | 2.83 [2.26, 3.34] | 2.93 [2.32, 3.42] | 0.141 | 2.68 [2.19, 3.19] | 2.63 [2.12, 3.12] | 0.128 |
| FBG, mmol/L | 5.01 [4.68, 5.59] | 5.26 [4.80, 6.12] | <0.001 | 4.97 [4.64, 5.43] | 5.20 [4.76, 5.82] | <0.001 | 5.13 [4.74, 5.79] | 5.29 [4.85, 6.31] | 0.001 |

**Abbreviations:** BMI, body mass index; SBP, systolic blood pressure; DBP, diastolic blood pressure; FBG, fasting blood glucose; TG, triglyceride; TG, total cholesterol; HDL-C, high-density lipoprotein cholesterol; LDL-C, low-density lipoprotein cholesterol.

**Table 2. Traits of enrolled female subjects at baseline in the different cohorts of metabolic obesity.**

| Variables | Total | MHNO | MHO | MUNO | MUO | P-value |
|---|---|---|---|---|---|---|
| No. of cases | 993 | 582 | 27 | 339 | 45 | – |
| Age, years | 57.00 [50.00, 63.00] | 54.00 [46.00, 60.00] | 55.00 [49.50, 60.50] | 61.00 [55.00, 68.00] | 61.00 [55.00, 66.00] | <0.001 |
| Smoking, n (%) | 12 (1.2) | 6 (1.0) | 0 (0.0) | 4 (1.2) | 2 (4.4) | 0.218 |
| Drinking, n (%) | 38 (3.8) | 22 (3.8) | 1 (3.7) | 12 (3.5) | 3 (6.7) | 0.785 |
| Diabetes, n (%) | 76 (7.7) | 9 (1.5) | 0 (0.0) | 56 (16.5) | 11 (24.4) | <0.001 |
| Hypertention, n (%) | 255 (25.7) | 79 (13.6) | 6 (22.2) | 143 (42.2) | 27 (60.0) | <0.001 |
| BMI, kg/m$^2$ | 22.83 [20.94, 24.89] | 22.00 [20.40, 23.78] | 29.38 [28.74, 30.67] | 23.71 [21.88, 24.97] | 29.53 [28.58, 30.61] | <0.001 |
| SBP, mmHg | 125.00 [114.00, 137.00] | 119.00 [110.00, 128.00] | 131.00 [120.00, 143.00] | 134.00 [124.00, 143.00] | 133.00 [123.00, 149.00] | <0.001 |
| DBP, mmHg | 77.00 [70.00, 84.00] | 75.00 [68.00, 81.00] | 81.00 [73.50, 87.50] | 80.00 [72.00, 86.00] | 84.00 [77.00, 89.00] | <0.001 |
| TC, mmol/L | 5.00 [4.33, 5.66] | 5.03 [4.38, 5.62] | 5.47 [4.66, 6.22] | 4.94 [4.28, 5.74] | 4.76 [4.25, 5.36] | 0.151 |
| TG, mmol/L | 1.25 [0.88, 1.75] | 1.01 [0.76, 1.34] | 1.29 [1.02, 1.56] | 1.82 [1.28, 2.44] | 1.74 [1.26, 2.56] | <0.001 |
| HDL-C, mmol/L | 1.34 [1.15, 1.57] | 1.44 [1.24, 1.69] | 1.32 [1.19, 1.48] | 1.19 [1.00, 1.40] | 1.20 [1.02, 1.33] | <0.001 |
| LDL-C, mmol/L | 2.86 [2.31, 3.36] | 2.91 [2.40, 3.33] | 3.44 [2.82, 3.76] | 2.69 [2.12, 3.38] | 2.80 [2.07, 3.28] | <0.001 |
| FBG, mmol/L | 5.03 [4.68, 5.59] | 4.84 [4.59, 5.15] | 5.09 [4.84, 5.31] | 5.58 [4.99, 6.32] | 5.65 [5.09, 6.46] | <0.001 |

**Abbreviations:** BMI, body mass index; SBP, systolic blood pressure; DBP, diastolic blood pressure; FBG, fasting blood glucose; TG, triglyceride; TG, total cholesterol; HDL-C, high-density lipoprotein cholesterol; LDL-C, low-density lipoprotein cholesterol.MHNO, metabolically healthy non-obesity; MHO, metabolically healthy obesity; MUNO, metabolically unhealthy non-obesity; MUO, metabolically unhealthy obesity.

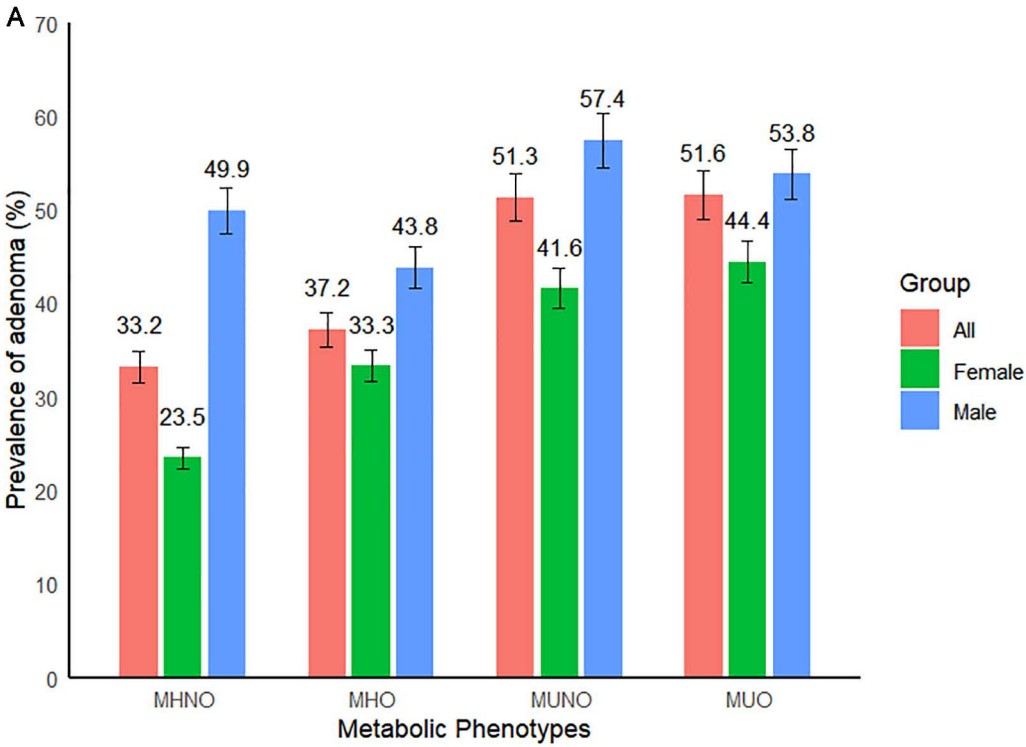

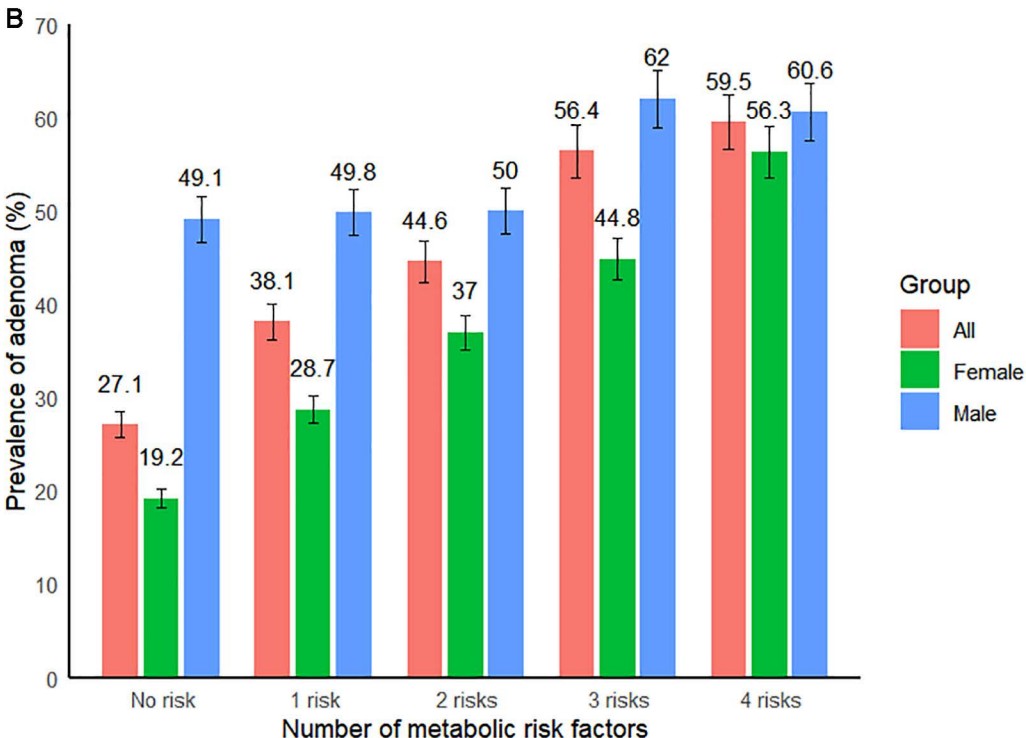

**Fig 1. Frequency of occurrence of CRA in the different cohorts of metabolic obesity according to the sex.** Frequency of occurrence of CRA in the Fig 1A different phenotypes of metabolic obesity, and Fig 1B according to the proportion of metabolic risk factors.

progressively with higher proportions of metabolic risk factors among female participants (P<0.05; Fig 1B). Metabolic parameters, such as SBP, FBG, and TG, were elevated in the metabolically unhealthy groups compared to the metabolically healthy groups; conversely, HDL-C and LDL-C exhibited an inverse association (all P<0.05). BMI levels were significantly greater in the obesity groups (MHO and MUO) compared to the normal weight groups (MHNO and MUNO). The prevalence of metabolic disorders, such as diabetes and hypertension, was greater in the metabolically unhealthy groups compared to the metabolically healthy groups (all P<0.05). Alcohol consumption and smoking exhibited no significant variations among the four cohorts of metabolic obesity.

The baseline characteristics of male participants (n = 1049) across distinct metabolic obesity phenotypes are presented in Table 3. The male individuals enrolled in the study had an average age of 57 years. The MHNO, MHO, MUNO, and MUO cohorts comprised 339 (32.3%), 16 (1.5%), 551 (52.5%), and 143 (13.6%) males, respectively. The frequency of occurrence of adenomas in these cohorts was 49.9%, 43.8%, 57.4%, and 53.8%, respectively (P<0.05; Fig 1A). Compared with participants with the MHNO and MHO phenotypes, those with the MUNO and MUO phenotypes displayed a significantly elevated occurrence of CRA (P<0.05). However, the occurrence of CRA showed no difference in the male participants with the proportion of metabolic risk factors (Fig 1B). The metabolic parameters DBP, FBG, and TG were elevated in the metabolically unhealthy groups compared to the metabolically healthy groups, whereas HDL-C exhibited an inverse association (all P<0.05). Moreover, BMI levels were significantly greater in the obesity groups (MHO and MUO) compared to the normal weight groups (MHNO and MUNO). Notably, the prevalence of metabolic disorders, such as diabetes and hypertension, was greater in the metabolically unhealthy groups compared to the metabolically healthy groups (all P<0.05). A significant difference in cigarette consumption was observed between the metabolically unhealthy and healthy groups.

## Sex-based differences in metabolic obesity phenotypes and colorectal adenoma

Exploratory analyses were performed in subgroups stratified by sex. Fig 2 illustrates the logistic analysis of CRA prevalence across different obesity phenotypes subgrouped by gender. The findings demonstrated that, irrespective of sex,

**Table 3. Traits of enrolled male subjects at baseline in the different cohorts of metabolic obesity.**

| Variables | Total | MHNO | MHO | MUNO | MUO | P-value |
|---|---|---|---|---|---|---|
| No. of cases | 1049 | 339 | 16 | 551 | 143 | – |
| Age, years | 57.00 [49.00, 64.00] | 56.00 [47.00, 64.00] | 52.50 [41.00, 64.00] | 59.00 [51.00, 66.00] | 53.00 [46.00, 59.00] | <0.001 |
| Smoking, n (%) | 434 (41.4) | 120 (35.4) | 8 (50.0) | 254 (46.1) | 52 (36.4) | 0.007 |
| Drinking, n (%) | 394 (37.6) | 111 (32.7) | 5 (31.2) | 219 (39.7) | 59 (41.3) | 0.134 |
| Diabetes, n (%) | 139 (13.3) | 3 (0.9) | 0 (0.0) | 113 (20.5) | 23 (16.1) | <0.001 |
| Hypertention, n (%) | 400 (38.1) | 60 (17.7) | 3 (18.8) | 258 (46.8) | 79 (55.2) | <0.001 |
| BMI, kg/m² | 24.61 [22.68, 26.73] | 23.34 [21.53, 24.81] | 29.24 [28.39, 30.65] | 24.61 [23.04, 26.04] | 29.39 [28.57, 30.47] | <0.001 |
| SBP, mmHg | 129.00 [118.00, 139.00] | 124.00 [115.00, 132.00] | 131.50 [121.25, 137.00] | 131.00 [120.50, 140.00] | 136.00 [123.50, 144.50] | <0.001 |
| DBP, mmHg | 82.00 [75.00, 89.00] | 79.00 [72.00, 85.00] | 81.00 [75.00, 89.25] | 82.00 [75.00, 90.00] | 88.00 [81.00, 94.00] | <0.001 |
| TC, mmol/L | 4.72 [4.11, 5.32] | 4.72 [4.20, 5.27] | 4.62 [4.12, 5.09] | 4.71 [3.98, 5.36] | 4.76 [4.14, 5.40] | 0.664 |
| TG, mmol/L | 1.64 [1.10, 2.56] | 1.13 [0.84, 1.46] | 1.16 [0.90, 1.58] | 2.03 [1.40, 2.93] | 2.34 [1.63, 3.22] | <0.001 |
| HDL-C, mmol/L | 1.09 [0.93, 1.27] | 1.23 [1.12, 1.44] | 1.24 [1.15, 1.39] | 1.00 [0.88, 1.19] | 0.96 [0.85, 1.10] | <0.001 |
| LDL-C, mmol/L | 2.67 [2.17, 3.15] | 2.79 [2.41, 3.17] | 2.52 [2.32, 3.01] | 2.54 [2.02, 3.13] | 2.66 [2.12, 3.17] | <0.001 |
| FBG, mmol/L | 5.22 [4.78, 6.06] | 4.93 [4.63, 5.28] | 4.92 [4.58, 5.09] | 5.57 [4.90, 6.52] | 5.55 [4.99, 6.40] | <0.001 |

**Abbreviations:** BMI, body mass index; SBP, systolic blood pressure; DBP, diastolic blood pressure; FBG, fasting blood glucose; TG, triglyceride; TG, total cholesterol; HDL-C, high-density lipoprotein cholesterol; LDL-C, low-density lipoprotein cholesterol.MHNO, metabolically healthy non-obesity; MHO, metabolically healthy obesity; MUNO, metabolically unhealthy non-obesity; MUO, metabolically unhealthy obesity.

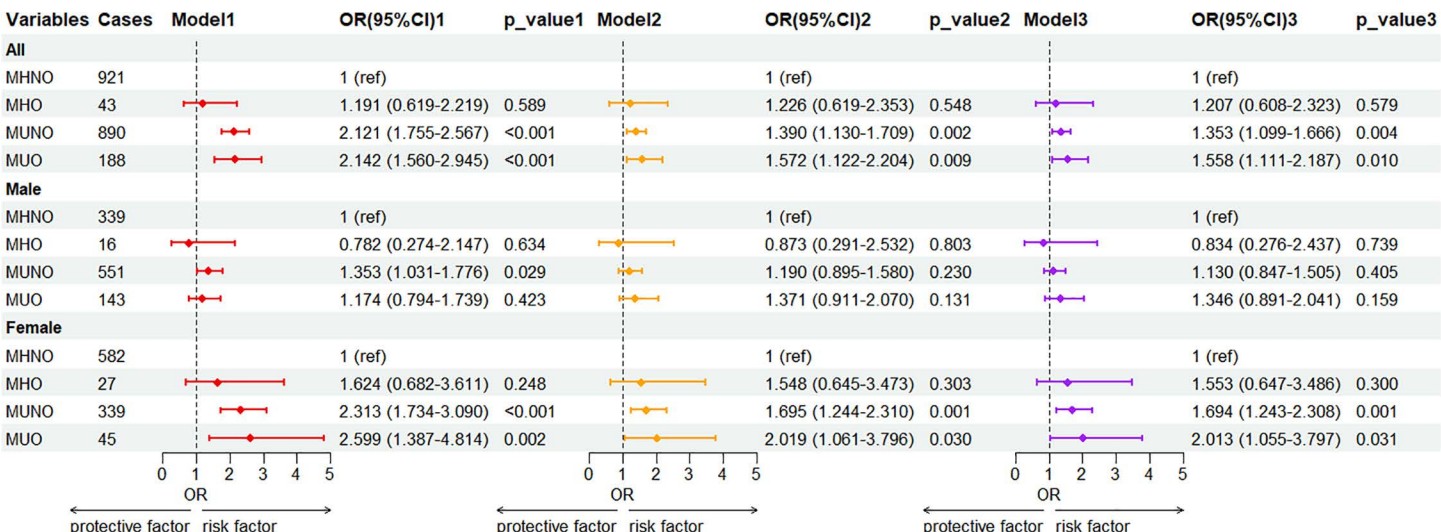

**Fig 2. Association between the metabolic obesity phenotypes and risks of developing CRA based on the sex.** Notes: Model 1: not adjusted. Model 2: adjusted for age and and sex. Model 3: adjustment for age, sex, smoking, and drinking. MHNO, metabolically healthy non-obesity; MHO, metabolically healthy obesity; MUNO, metabolically unhealthy non-obesity; MUO, metabolically unhealthy obesity.P-value < 0.01 was used as the significance threshold within subgroups.

the individuals in the MUNO and MUO cohorts exhibited elevated risks of developing CRA compared to those in the MHNO cohort (P < 0.05). Nonetheless, the MHO cohort showed a similar rate of CRA occurrence as the MHNO group. After adjusted for age, sex, smoking and alcohol intake, the adjusted odds ratios (ORs) (95% CI) for the frequency of occurrence of CRA in the MUNO and MUO cohorts were determined to be 1,353 (1.099–1.666) and 1.558 (1.111–2.187) compared to the MHNO phenotype, respectively. After adjusting for age, smoking, and alcohol intake, the female participants with the MUNO phenotype (OR 1.694; 95% CI 1.243–3.308, p < 0.01) exhibited significantly elevated risks of CRA compared to those in the MHNO group. Among the male participants, after adjusting for age, smoking, and alcohol intake, no significant differences were found across the four groups of metabolic health phenotypes.

## Age-based differences in metabolic obesity phenotypes and colorectal adenoma

Fig 3 presents the relationships between the different metabolic obesity phenotypes and the prevalence of CRA stratified by age. Regardless of age, the MUNO and MUO phenotypes were identified as risk factors for developing CRA, unlike the MHNO phenotype (p < 0.05). After adjusting for age, sex, smoking, and alcohol intake, the adjusted ORs (95% CI) for the prevalence of CRA in the MUNO and MUO cohorts were determined to be 1,353 (1.099–1.666) and 1.558 (1.111–2.187) compared to that of the MHNO phenotype, respectively. Among participants aged less than 60 years, after adjusting for sex, smoking, and alcohol consumption, the MUNO group (OR: 1.648; 95% CI: 1.252–2.169, p < 0.01) demonstrated significantly elevated risks of developing CRA compared to the MHNO cohorts. In contrast, MHO individuals and MUO individuals showed no significantly increased risk of CRA compared with MHNO individuals. Among the participants aged above 60 years, after adjusting for sex, smoking status, and alcohol consumption, the individuals in the MUO cohort exhibited the greatest OR of 2.301 (95% CI: 1.252–4.348, p < 0.01) among all the cohorts.

## Sensitivity analysis

To test the robustness of our findings, we redefined obesity using the World Health Organization (WHO) recommendation for Asian populations (BMI ≥ 25 kg/m²) instead of the Chinese criteria (BMI ≥ 28 kg/m²). All other statistical models

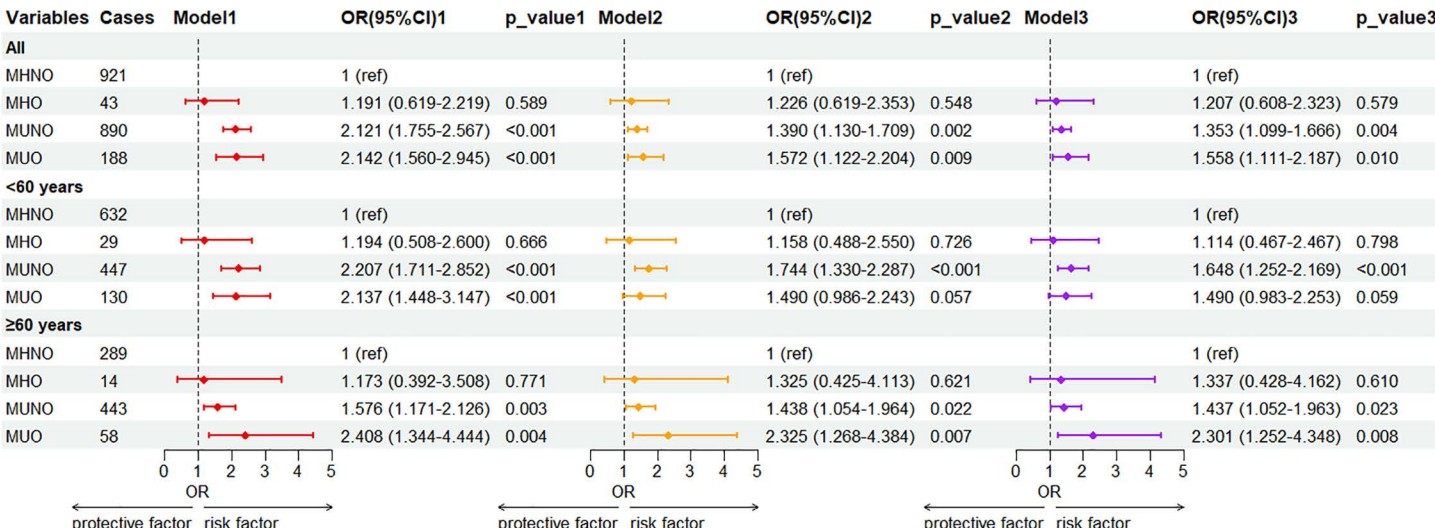

**Fig 3. Association between the metabolic obesity phenotypes and risks of developing CRA based on the age.** Notes: Model 1: not adjusted. Model 2: adjustment for age, and sex. Model 3: adjustment for age, sex, smoking, and drinking.MHNO, metabolically healthy non-obesity; MHO, metabolically healthy obesity; MUNO, metabolically unhealthy non-obesity; MUO, metabolically unhealthy obesity.P-value < 0.01 was used as the significance threshold within subgroups.

remained identical to the primary analysis(S1 Table).The metabolically unhealthy groups (MUNO and MUO) also had the highest risk of CRA compared with the MHNO group.

## Discussion

This cross-sectional study analyzed the association between different phenotypes of metabolic obesity and the incidence of CRA. The results revealed that the MUNO and MUO phenotypes were risk factors for CRA development compared to the MHNO phenotype. Collectively, the findings demonstrate that metabolically unhealthy status is more closely associated with CRA pathogenesis than BMI-defined obesity, which highlights the importance of focusing on metabolic health as a preventive measure against CRA. In the gender subgroup analysis, the occurrence of MUNO or MUO was associated with an elevated incidence of CRA in females. In contrast, no significant difference was found in the incidence of CRA among the four groups in males. The age subgroup analysis showed that the MUNO phenotype was a risk factor for CRA among participants under 60 years, while both the MUNO and MUO phenotypes were risk factors in participants over 60 years. It is important to note that while there is no universally accepted definition of "metabolic health", the criteria used in this study (based on Adult Treatment Panel III criteria and the China Guidelines for Type 2 Diabetes) have been widely adopted in epidemiological research for their clinical practicality. This definition facilitates comparison with a substantial body of existing literature. However, we acknowledge that alternative definitions incorporating insulin resistance or waist circumference might capture different aspects of metabolic dysfunction, and findings may vary accordingly. However, the chosen definition was selected for its feasibility within large-scale health examination datasets.

This study conducted comparative analyses across multiple metabolic phenotypes, revealing statistically significant differences in adenoma detection rates (P < 0.05). Compared with the MHNO group, the MHO, MUNO, and MUO groups all demonstrated elevated adenoma detection rates. However, across the entire enrolled population, the results of our logistic analyses showed adjusted ORs greater than 1 in the MUO and MUNO groups compared to the MHNO group, which suggests that metabolic dysfunction may play a more important pathogenetic role in the formation of CRA compared to

obesity. The precise mechanism underlying this link remains unknown. However, some studies have revealed that hyperglycemia, dyslipidemia, and hypertension are strongly associated with CRA, which is consistent with our findings.

A previous study investigated the connection between a high intake of simple sugars and the prevalence of CRA [20]. The results indicated an association between an elevated risk of rectal adenomas and a higher intake of simple sugars, indicating that tighter glycemic management should be given priority for younger populations [21]. The potential mechanisms are as follows. Firstly, excessive glucose concentrations directly damage human endothelial cells' DNA [20], which can result in alterations in intestinal mucosa permeability and compromise the integrity of the gut barrier [22]. Secondly, some intestinal microbial compounds, including lipopolysaccharides, can cross the intestinal barrier and induce inflammatory processes that promote cell survival and proliferation, resulting in CRA [23,24]. Thirdly, dysregulation of the insulin/IGF-1 signaling pathway may be involved. Insulin can act as an anabolic and oncogene, either directly or indirectly by regulating IGF-1 [25]. This may accelerate CRA growth and the spread of colorectal cancer cells. Fourthly, the Warburg effect: under persistent hyperglycemic conditions, cancer cells gain a metabolic advantage by preferentially utilizing glucose and transforming glycolytic intermediates into biosynthetic pathways [26].

As endogenous mediators, lipids play a crucial role in many physiological processes, including membrane trafficking, cell signaling, apoptosis, and cell proliferation [27]. Hence, dyslipidemia impacts the microenvironment, resulting in neovascularization, cell proliferation, and DNA damage [28]. The majority of studies have revealed that elevated HDL levels are inversely correlated with the formation of CRA, whereas higher blood TG levels are substantially linked to an increased risk of CRA [29,30]. Dyslipidemia may contribute to the development of insulin resistance and hyperinsulinemia, prevent apoptosis through its interaction with the insulin-like growth factor 1 (IGF-1) receptor, and encourage the growth of colorectal cells, which can result in the formation of CRA and the initiation of cancer [31]. Furthermore, lipid disorders have been shown to alter the bile acid cycle, which may raise gut bile acid levels [32] and and promote dysbiosis and carcinogen production, leading to an increased risk of CRA [33]. The release of inflammatory cytokines is also associated with lipid abnormalities. Conversely, the release of tumor necrosis factor and anti-inflammatory cytokines is associated with a reduction in the survival of tumor cells, which may facilitate the growth of CRA and cancerous cells [34,35].

The existing epidemiological evidence on the association between CRA and hypertension is inconclusive, with only a few studies reporting positive results. Kaneko et al. analyzed data from the National Health Claims Database and demonstrated that stage 2 hypertension and elevated systolic and diastolic blood pressure were associated with an increased risk of developing CRC [36]. In a prior study, hypertension model rats developed aberrant crypt foci (ACF) and colonic preoneoplastic lesions by azoxymethane exposure more quickly than normotensive rats. Oxidative stress and exacerbation of colonic mucosal inflammation were observed in hypertensive rats. Furthermore, the administration of an angiotensin-converting enzyme inhibitor resulted in a notable reduction in the total number and size of ACF compared to the untreated control group. These findings suggest that reducing inflammation and oxidative stress may inhibit the development of ACF. The renin-angiotensin-aldosterone system may be a key factor in the development of CRA and CRC. Considering that angiotensin is a growth factor that encourages angiogenesis and cell proliferation, the renin-angiotensin system plays a significant role in controlling blood pressure and cell growth [37]. However, further research is required to fully elucidate the potential pathways and associations between hypertension and recurrent CRA.

Hyperglycemia, hypertension, and dyslipidemia share the pathogenic mechanisms of insulin resistance and inflammation and are closely related. Given the above results, a metabolically unhealthy phenotype could be a more significant contributing factor to the formation of CRA. Furthermore, the mechanisms linking metabolic abnormalities to CRA may exhibit distinct characteristics in Asian, particularly Chinese, which should be considered when interpreting our findings. Epidemiological and physiological studies indicate that Chinese individuals are susceptible to developing metabolic syndrome and significant visceral adiposity at lower BMI thresholds compared to Caucasian populations [38,39]. This "lean but metabolically unhealthy" phenotype may suggests that the pathogenic impact of dysmetabolism on the colorectum might be initiated earlier in the disease course and be less dependent on overall obesity as defined by BMI alone.

Moreover, dietary patterns prevalent in China, characterized by high consumption of refined carbohydrates and low intake of dietary fiber [19], may exacerbate postprandial hyperglycemia, insulin resistance, and systemic inflammation—key drivers implicated in CRA pathogenesis. These dietary habits could potentiate the mechanisms described above, such as gut barrier dysfunction and dysbiosis, in a population-specific manner [40]. Additionally, genetic factors and environmental factors in different regions play distinct roles in colorectal cancer development [41]. Therefore, while common pathological mechanisms such as hyperglycemia, dyslipidemia, and hypertension are associated, their relative contributions, interactions, and clinical manifestations in colorectal cancer may differ among Chinese populations. Future additional research integrating population-specific biomarkers with diverse dietary pattern data is required to elucidate these relevant pathways and develop more precise and effective clinical prevention strategies.

Confounding factors were adjusted for, and no independent correlation was found between the MHO phenotype and CRA risk. Our findings corroborate the results of Kim et al. [42], which indicated that BMI-defined MHO individuals did not exhibit significantly increased CRA risk. Notably, longitudinal follow-up data from the Meigs [43] research team revealed no significant associations between the MHO phenotype and either cardiovascular endpoints or diabetes incidence. Importantly, 42.1% of baseline MHO individuals experienced deterioration in metabolic parameters during a 10-year follow-up period [44], ultimately progressing to the MUO phenotype. Collectively, this evidence reconceptualizes MHO as a dynamic phenotype in the metabolic compensation phase rather than a stable clinical subtype.The current conclusion that "no independent association between MHO phenotype and CRA risk" may be limited by the duration of phenotypic observation. Future prospective cohort studies are needed to track the long-term impact of phenotypic conversion on CRA occurrence, thereby avoiding conclusions biased by short-term observation. Consequently, implementing lifestyle interventions prior to phenotypic transition is recommended to reduce the incidence of metabolic abnormalities.

To investigate the impact of gender on the relationship between CRA and the metabolic obesity phenotype, the participants were stratified by sex. Our analysis revealed that among females, metabolically unhealthy individuals also appeared to have a greater propensity for CRA than those with healthy metabolism. Obesity status showed minimal to no influence on adenoma occurrence. Subsequent multiple logistic analyses using the adjusted model revealed that females with MUNO phenotypes had significantly elevated adjusted OR for CRA.Interestingly, we did not observe a significant association between metabolic phenotypes and CRA in the male subgroup. This could suggest a genuine biological difference; however, it is also important to note that the sample sizes for some phenotypes (particularly MHO and MUO) in the male subgroup were relatively small, which may have limited our statistical power to detect existing associations, leaving the possibility of false negatives. Therefore, the conclusion that no significant differences were found among male subgroups should be interpreted with caution and requires further validation in larger-scale studies to explore the biological mechanisms underlying gender differences. A previous study investigated a colon cancer mouse model; male mice exhibited a significantly higher incidence of CRC than female mice [45]. Additionally, administration of 17 β-estradiol (E2) during the inflammatory phase was shown to inhibit CRC development, highlighting the protective effect of estrogen in inflammatory responses [46]. Moreover, estrogen plays a regulatory role in insulin sensitivity and lipid metabolism [47]. Furthermore, females generally adopt healthier lifestyles and dietary practices compared to males [47]. These findings suggest that gender disparities in CRA may arise from a combination of biological and behavioral factors. The observed gender differences in CRA incidence highlight the importance of considering gender-specific approaches in prevention, screening, and treatment strategies to improve outcomes. Cheng's study [16] indicated an increased incidence of colorectal polyps in postmenopausal women, which may be attributed to lower estrogen levels. Future studies should increase the sample size and perform stratified analyses based on menopausal status and estrogen levels to explore the correlation between metabolic obesity phenotype, estrogen levels, and CRA.

Furthermore, this study investigated the relationship between metabolic obesity phenotypes and the occurrence of CRA in different age groups. Our analysis revealed that among participants under 60 years, individuals with the MUNO phenotype exhibited higher risks of developing CRA compared to the individuals with the MHNO phenotype. Among individuals

over 60 years, the MUO phenotype was identified as risk factors for CRA. These findings indicate that age affects the relationship between metabolic health status and colorectal adenoma risk, with metabolic abnormalities alone raising risk in younger populations, while obesity and metabolic abnormalities in combination may increase risk in older populations. This research emphasizes the necessity of optimizing colonoscopy screening indications in high-risk groups while developing age-specific metabolic management methods in clinical practice.

Nevertheless, the limitations of the present study should be acknowledged. First, The cross-sectional study design of this research can not determine the causal relationship between the metabolic obesity phenotype and CRA. The exposure and outcome were assessed simultaneously, making it impossible to determine temporality. Furthermore, MHO is recognized as a potentially transient state, with a significant proportion of individuals progressing to MUO over time. Our study provides only a snapshot in time, and we could not ascertain the duration for which individuals maintained their MHO phenotype or their potential for future transition. Therefore, the lack of association observed for MHO in our study might reflect insufficient duration of exposure rather than a true absence of risk. Future prospective cohort studies are needed to track the long-term impact of phenotypic conversion on CRA risk, thereby avoiding conclusions biased by short-term observation. Second, obesity is defined only by BMI, Supplementary indicators reflecting central obesity, such as waist circumference and body fat percentage, were not included.However, some studies have also explored the association between central obesity and metabolic abnormalities with intestinal lesions. Future research should integrate body composition analysis to refine the relationship between different obesity subtypes and CRA. Third, this study only accounted for smoking and alcohol consumption as confounding factors, key lifestyle factors such as dietary patterns and physical activity were not considered, and these confounding factors were not accounted for. These factors may have influenced the risk assessment results of this study. Moreover, adenoma detection rates may also have varied throughout the study period due to differences in skills between gastroenterologists.

## Conclusion

This is the first study in China to investigate the correlation between metabolic health phenotypes and CRA. The present study confirmed that MUO and MUNO are risk factors for CRA compared to MHNO. This seems to prove that metabolism may have a greater impact on CRA than obesity, which serves as a reminder to focus on people with metabolic disorders during clinical screening. Second, our investigation demonstrated gender-specific and age-specific disparities in the linkage between metabolic obesity phenotypes and CRA development.

## Supporting information

**S1 Table. Sensitivity analysis.** Notes: Model 1: not adjusted. Model 2: adjustment for age, and sex. Model 3: adjustment for age, sex, smoking, and drinking.
(DOC)

**S2 Data. Raw data underlying the analyses presented in this study.**
(DOCX)

## Acknowledgments

The authors thank all patients and research staffs who participated in this study, and also gratefully thank the nurses for their help in the process of data collection.

## Author contributions

**Data curation:** Li Lin, Yuhan Ying.

**Formal analysis:** Xiaoling Lv.

**Methodology:** Qin Zhu.

**Supervision:** Long Shu, Qin Zhu.

**Writing – original draft:** Li Lin.

**Writing – review & editing:** Li Lin.

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
