## [Decision Letter · Decision Letter 0]

26 Nov 2025

Dear Dr. Zhu,

Thank you for submitting your manuscript to PLOS One. Firstly, we would like to apologize for the delay in processing your manuscript. It has been exceptionally difficult to secure reviewers to evaluate your study. We have now received one completed review, which is available below. The reviewer has raised significant scientific concerns about the study that need to be addressed in a revision.

We look forward to receiving your revised manuscript.

Kind regards,

Miquel Vall-llosera Camps

Senior Staff Editor

PLOS ONE

Journal Requirements:

4. Please note that your Data Availability Statement is currently missing or the DOI/accession number of each dataset OR a direct link to access each database. If your manuscript is accepted for publication, you will be asked to provide these details on a very short timeline. We therefore suggest that you provide this information now, though we will not hold up the peer review process if you are unable.

Additional Editor Comments:

Please note that we have only been able to secure a single reviewer to assess your manuscript. We are issuing a decision on your manuscript at this point to prevent further delays in the evaluation of your manuscript. Please be aware that the editor who handles your revised manuscript might find it necessary to invite additional reviewers to assess this work once the revised manuscript is submitted. However, we will aim to proceed on the basis of this single review if possible.

Reviewer's Responses to Questions

**Comments to the Author**

1. Is the manuscript technically sound, and do the data support the conclusions?

Reviewer #1: Yes

2. Has the statistical analysis been performed appropriately and rigorously?

Reviewer #1: Yes

3. Have the authors made all data underlying the findings in their manuscript fully available?

Reviewer #1: Yes

4. Is the manuscript presented in an intelligible fashion and written in standard English?

Reviewer #1: Yes

Reviewer #1: This study identified metabolically unhealthy status (irrespective of obesity) as a significant risk factor for CRA, offering valuable clinical insights. It refines the conventional broad concept of "obesity" into four distinct phenotypes, underscoring that metabolic health is more critical than body mass index (BMI) alone. The conclusions suggest that clinical screening for CRA should prioritize metabolically unhealthy individuals and emphasize the importance of managing metabolic syndrome (e.g., hypertension, hyperglycemia, dyslipidemia), which may directly benefit reducing colorectal cancer risk, beyond preventing cardiovascular diseases.

My comments as follows:

1.The study used a cross-sectional design, measuring both exposure (metabolic obesity phenotype) and outcome (presence of CRA) at a single time point. This approach cannot establish temporal causality. MHO is known to be a transient state, with many individuals progressing to MUO over time. Since the study only captured a snapshot, it remains unknown how long the MHO individuals had maintained this phenotype or whether they might transition in the future. Thus, the conclusion that "MHO is not associated with CRA risk" may be short-sighted, potentially reflecting insufficient exposure duration rather than a true lack of association. A prospective cohort study would be the ideal design to address this, though this remains an irreparable limitation of the current study. This point should be discussed more thoroughly in the manuscript.

2.There is no universally accepted definition for "metabolically healthy" and "metabolically unhealthy" (e.g., some use ATP-III metabolic syndrome criteria, others use HOMA-IR for insulin resistance). Varying definitions can significantly alter the classification of MHO/MUNO individuals, directly affecting the stability and comparability of the results. The manuscript must transparently report the specific criteria used (including BMI cutoffs, metabolic indicators, and their thresholds). Where possible, sensitivity analyses using multiple established definitions should be conducted to test the robustness of the findings.

3.In the male subgroup, no significant differences were observed among the four phenotypes, contradicting the results in the overall and female populations, may be due to: (a) a genuine biological effect—metabolic abnormalities may influence CRA risk differently in men (e.g., via interaction with sex hormones); or (b) insufficient sample size—particularly in the MHO and MUO male subgroups, leading to low statistical power and an inability to detect actual differences (i.e., a false negative). Therefore, it is crucial to report the exact sample sizes (n) for each subgroup. If samples are small, the authors should explicitly state that the negative results may be due to low power and interpret them cautiously. Larger studies are needed to validate these sex-specific differences.

4.The study conducted numerous statistical comparisons (across four phenotypes in the overall population, males, females, and age groups <60/≥60 years), but did not mention whether correction for multiple testing (e.g., Bonferroni correction) was applied. When performing extensive subgroup analyses, a stricter significance level (e.g., p < 0.01) or appropriate multiplicity adjustment should be used, and uncorrected results must be interpreted with caution.

**Do you want your identity to be public for this peer review?** For information about this choice, including consent withdrawal, please see our Privacy Policy

Reviewer #1: No

---

## [Author Response · Author response to Decision Letter 1]

2 Dec 2025

Dear Reviewer,

We appreciate the opportunity to revise our manuscript titled “Association between different metabolic obesity phenotypes and colorectal adenoma” and are grateful for the insightful comments provided by the peer-reviewer. Those comments are all valuable and very helpful for revising and improving our paper, as well as the important guiding significance to our researches. We have tried our best to make all the revisions clear, and we hope that the revised manuscript meets the requirements for publication. In the following, we have provided a point-by-point response addressing each issue raised in the peer-review report:

Comment 1: The study used a cross-sectional design, measuring both exposure (metabolic obesity phenotype) and outcome (presence of CRA) at a single time point. This approach cannot establish temporal causality. MHO is known to be a transient state, with many individuals progressing to MUO over time. Since the study only captured a snapshot, it remains unknown how long the MHO individuals had maintained this phenotype or whether they might transition in the future. Thus, the conclusion that "MHO is not associated with CRA risk" may be short-sighted, potentially reflecting insufficient exposure duration rather than a true lack of association. A prospective cohort study would be the ideal design to address this, though this remains an irreparable limitation of the current study. This point should be discussed more thoroughly in the manuscript.

Response: We agree with the reviewer that the cross-sectional design is a limitation for establishing causality and that the transient nature of the MHO phenotype is an important consideration. As suggested, we have now added a detailed discussion on this limitation in the Discussion section of the manuscript(Line 385 to Line 396).

Comment 2: There is no universally accepted definition for "metabolically healthy" and "metabolically unhealthy" (e.g., some use ATP-III metabolic syndrome criteria, others use HOMA-IR for insulin resistance). Varying definitions can significantly alter the classification of MHO/MUNO individuals, directly affecting the stability and comparability of the results. The manuscript must transparently report the specific criteria used (including BMI cutoffs, metabolic indicators, and their thresholds). Where possible, sensitivity analyses using multiple established definitions should be conducted to test the robustness of the findings.

Response: We sincerely thank the reviewer for this critical and constructive comment. We agree that the lack of a universal definition is a key consideration. We have now explicitly and transparently detailed the specific criteria used to define metabolic health status and obesity in the Methods section(Line 121 to Line 132). This includes the exact components, cut-off values, and the reference standards we adopted; While we did not measure leptin or fasting insulin levels, which precluded a sensitivity analysis using HOMA-IR, we fully recognized the importance of testing the robustness of our findings. Therefore, we conducted an alternative sensitivity analyses using the data available to us: We altered the definition of obesity. In addition to the primary definition of BMI ≥ 28 kg/m² (Chinese criteria), we tested the World Health Organization (WHO) standard for Asian populations (BMI ≥ 25 kg/m²) to define obesity(Line 247 to Line 253).

We are pleased to report that the results of both sensitivity analyses were consistent with our primary findings. The core conclusion—that metabolically unhealthy phenotypes, irrespective of obesity status, are associated with higher odds of CRA—remained robust. These new results have been added to the Results sections of the revised manuscript (see Supplementary Tables S1).

Comment 3: In the male subgroup, no significant differences were observed among the four phenotypes, contradicting the results in the overall and female populations, may be due to: (a) a genuine biological effect—metabolic abnormalities may influence CRA risk differently in men (e.g., via interaction with sex hormones); or (b) insufficient sample size—particularly in the MHO and MUO male subgroups, leading to low statistical power and an inability to detect actual differences (i.e., a false negative). Therefore, it is crucial to report the exact sample sizes (n) for each subgroup. If samples are small, the authors should explicitly state that the negative results may be due to low power and interpret them cautiously. Larger studies are needed to validate these sex-specific differences.

Response: We agree that the sample size in subgroups is a key issue. We have now: First, clearly reported the exact sample sizes (n) for each phenotype within the male and female subgroups in the corresponding Figures(Figure 2 and Figure 3). Second, added a statement in the Results and Discussion sections to caution the interpretation of the non-significant finding in the male subgroup, explicitly acknowledging the possibility of a false negative due to limited sample size and lower statistical power(Line 351 to Line 357).

Comment 4: The study conducted numerous statistical comparisons (across four phenotypes in the overall population, males, females, and age groups <60/≥60 years), but did not mention whether correction for multiple testing (e.g., Bonferroni correction) was applied. When performing extensive subgroup analyses, a stricter significance level (e.g., p < 0.01) or appropriate multiplicity adjustment should be used, and uncorrected results must be interpreted with caution.

Response: We sincerely thank the reviewer for raising this crucial methodological point. We agree that appropriate control for multiple testing is essential. Therefore, in the revised manuscript, the primary analysis in the overall population, which tests our main hypothesis, is interpreted at the conventional significance level of P < 0.05. For the extensive subgroup analyses (by sex and age), we applied a more stringent significance level of P < 0.01 for all tests within these subgroups, as detailed in the 'Statistical Analysis' section of the Methods(Line 151 to Line 155). This approach balances the need to mitigate the inflation of Type I error while maintaining reasonable power to detect potential effects in these secondary analyses.

We have explicitly stated this strategy in the methods, noted it in the results (including table footnotes) and discussed the cautious interpretation of subgroup findings in the context of multiple testing in the Discussion section(Line 351 to Line 357). We believe this significantly strengthens the statistical rigor and interpretation of our findings.

We appreciate your time and consideration, and we anticipate your favorable response at your earliest convenience.

Sincerely yours,

Dr. Qin Zhu

Professor

Department of Gastroenterology,

Zhejiang Hospital

HangZhou 310013, P. R. China

Tel: (86)-17706413537

Email: zhuqin-1@163.com

---

## [Decision Letter · Decision Letter 1]

2 Feb 2026

Dear Dr. Zhu,

The study clinically relevant, methodologically sound, and within the scope of the journal, but requests clearer discussion of limitations, subgroup interpretation, and conceptual definitions.

The required revisions are primarily related to strengthening the Discussion section and improving transparency of subgroup reporting, and do not affect the validity of the main findings. The authors are therefore invited to revise the manuscript accordingly.

plosone@plos.org . A letter that responds to each point raised by the academic editor and reviewer(s). You should upload this letter as a separate file labeled 'Response to Reviewers'.A marked-up copy of your manuscript that highlights changes made to the original version. You should upload this as a separate file labeled 'Revised Manuscript with Track Changes'.An unmarked version of your revised paper without tracked changes. You should upload this as a separate file labeled 'Manuscript'.

We look forward to receiving your revised manuscript.

Kind regards,

Muhammad Shahzad Aslam, Ph.D.,M.Phil., Pharm-D

Academic Editor

PLOS One

Journal Requirements:

**Additional Editor Comments:**

Study Design and Limitations

Emphasize more clearly in the Discussion that the cross-sectional design does not allow causal inference.

Note that the metabolically healthy obesity (MHO) phenotype may be transient and that long-term cohort studies are needed to evaluate phenotype transitions and CRA risk.

Acknowledge the limitation of defining obesity using BMI alone, and highlight the absence of central obesity indicators (e.g., waist circumference, body fat percentage). Indicate that future studies should integrate body composition measures.

Subgroup Analyses

Clearly report subgroup sample sizes (especially in male phenotypes) in the Results tables.

In the Discussion, interpret non-significant findings in males cautiously and acknowledge the possibility of limited statistical power and potential false-negative results.

Confounding Factors

Expand the limitations to mention unmeasured lifestyle confounders such as diet and physical activity, and discuss their potential impact on the observed associations.

Mechanistic Discussion

Strengthen the mechanism section by briefly discussing metabolic features that may be particularly relevant in the Chinese population, supported by existing literature.

Definition of Metabolic Health

Add a short comparison between your definition of metabolic health and other commonly used definitions.

Justify the rationale for selecting the current classification to enhance interpretability and comparability with other studies.

These revisions are mainly explanatory and conceptual and do not require additional analyses. Please revise accordingly and provide a point-by-point response.

Reviewers' comments:

Reviewer's Responses to Questions

**Comments to the Author**

Reviewer #1: All comments have been addressed

Reviewer #2: (No Response)

2. Is the manuscript technically sound, and do the data support the conclusions?

Reviewer #1: Yes

Reviewer #2: Yes

3. Has the statistical analysis been performed appropriately and rigorously?

Reviewer #1: Yes

Reviewer #2: Yes

4. Have the authors made all data underlying the findings in their manuscript fully available?

Reviewer #1: Yes

Reviewer #2: Yes

5. Is the manuscript presented in an intelligible fashion and written in standard English?

Reviewer #1: Yes

Reviewer #2: Yes

Reviewer #1: This is my second review of the article and all the comments I previously made have been addressed by the authors.

No additional issues have been identified. The revisions made by the authors are greatly appreciated.

In particular, the author analyzed the limitations of the current study in the discussion section and expressed the hope that further research will be conducted to address the existing issues.

Reviewer #2: Dear author:

This study focuses on the association between metabolic obesity phenotypes and colorectal adenoma (CRA), a topic closely aligned with clinical needs. As the main precancerous lesion of colorectal cancer (CRC), identifying the risk phenotypes of CRA is of great guiding value for early screening. The study enrolled 2042 Chinese participants, verified the robustness of results through stratified analyses (by sex and age) and sensitivity analysis, and featured a clear design logic with sufficient data support. The core conclusion (metabolically unhealthy phenotypes are risk factors for CRA) holds clinical reference significance, and the overall work complies with the publication scope and academic standards of PLOS ONE.

Main Strengths

1. Practical research perspective: Breaks through the traditional research on the association between simple obesity and CRA, refining into four metabolic obesity phenotypes. It reveals that "metabolic health status" has a more significant impact on CRA risk than BMI, providing a new target for clinical screening.

2. Comprehensive analysis dimensions: Incorporates stratified analyses by sex and age, clarifying risk differences among different populations, making the conclusions more targeted.

3. Relatively rigorous methodology: Adopts multivariate logistic regression to adjust for confounding factors, and verifies the robustness of core conclusions through sensitivity analysis using two BMI standards.

4. Transparent data reporting: Detailedly discloses the definition standards of metabolic phenotypes, statistical methods, and subgroup sample sizes, facilitating readers' reproduction and verification.

Issues to Be Improved and Suggestions

1 In-depth Discussion on Limitations of Study Design

1. The cross-sectional design cannot establish a temporal causal relationship between metabolic obesity phenotypes and CRA. Additionally, the MHO phenotype is transient, and the existing conclusion that "MHO is not associated with CRA risk" may be limited by the duration of phenotype observation. It is recommended to further clarify in the discussion that prospective cohort studies are needed in the future to track the long-term impact of phenotype transition on CRA occurrence, avoiding conclusion bias caused by short-term observation.

2. Obesity is only defined by BMI, without including complementary indicators reflecting central obesity such as waist circumference and body fat percentage. However, central obesity is more closely associated with metabolic abnormalities and intestinal lesions. It is suggested to supplement the discussion on this limitation and point out that future studies should integrate body composition analysis to refine the impact of different obesity subtypes.

2 Supplementary Explanations for Subgroup Analyses

1. The sample sizes of some phenotypes in the male subgroup are relatively small, leading to insufficient statistical power which may mask true associations. It is recommended to clearly mark the sample size of each subgroup in the result tables, and emphasize in the discussion that the "no significant difference in males" should be interpreted with caution. The false negative result cannot be excluded due to limited sample size, and larger-scale studies are required to verify the biological mechanisms underlying sex differences.

3 Improvement of Confounding Factor Control and Mechanism Discussion

1. The current confounding factors only include smoking and alcohol consumption, without incorporating key lifestyle factors such as dietary structure and physical activity, which may simultaneously affect metabolic status and CRA occurrence. It is recommended to supplement the discussion on the potential impact of this omission on the results or explain it in the limitations section.

2. The mechanism section only generally describes the role of hyperglycemia and dyslipidemia, lacking targeted analysis of the metabolic characteristics of the Chinese population. It is suggested to briefly supplement the specificity of relevant mechanisms in the Chinese population by referencing existing literature to enhance the depth of the discussion.

4 Supplementary Explanations for Definitions and Classifications

Although the definition standards of metabolic health have been clarified, there is no universal standard for "metabolic health". It is recommended to compare the definition used in this study with other mainstream definitions in the discussion, explain the rationality of the selected standard, and enhance the comparability of results.

Conclusion and Publication Recommendation

This study has clear clinical value and academic significance, with reliable core conclusions and basically rigorous methodology. The aforementioned issues are mostly related to in-depth discussion of limitations and supplementary explanations, which do not affect the validity of the core conclusions.

Recommendation: Accept after minor revisions. Please the authors supplement and improve the discussion section in response to the above suggestions, clearly mark subgroup sample sizes, refine limitations and future research directions, and further enhance the completeness and academic rigor of the manuscript.

**Do you want your identity to be public for this peer review?** For information about this choice, including consent withdrawal, please see our Privacy Policy

Reviewer #1: No

Reviewer #2: No

---

## [Author Response · Author response to Decision Letter 2]

4 Feb 2026

Dear Editor and Reviewers,

Thank you very much for your comments regarding our manuscript entitled “Association between different metabolic obesity phenotypes and colorectal adenoma” We are truly grateful to your valuable comments and thoughtful suggestions on how to improve our manuscript. The valuable comments have provided important guidance for our research and the revision of our manuscript. Based on these comments and suggestions, we have made appropriate modifications on the original manuscript, we hope that the revised manuscript meets the requirements for publication. In the following, we have provided a point-by-point response addressing each issue raised by the academic editor and reviewer:

Responds to academic editor’ comments:

Comment 1: Emphasize more clearly in the Discussion that the cross-sectional design does not allow causal inference.

Response: We agree with the editor that the cross-sectional design is a limitation for establishing causality. As suggested, we have now explicitly stated that due to the cross-sectional nature of our study, the observed associations cannot be interpreted as causal relationships in the Limitation section of the manuscript(Line 462 to Line 464).

Comment 2: Note that the metabolically healthy obesity (MHO) phenotype may be transient and that long-term cohort studies are needed to evaluate phenotype transitions and CRA risk.

Response: We sincerely thank the editor for this critical and constructive comment. We added a paragraph in the Discussion section noting that the MHO phenotype is potentially dynamic and may transition to metabolically unhealthy states over time. We now emphasize that prospective cohort studies are required to track these transitions and their longitudinal relationship with CRA risk(Line 413 to Line 417). And in the Limitation section, we also noted that this study have conclusion bias due to its short observation period, and emphasized the need for future long-term prospective research(Line 471 to Line 473).

Comment 3: Acknowledge the limitation of defining obesity using BMI alone, and highlight the absence of central obesity indicators (e.g., waist circumference, body fat percentage). Indicate that future studies should integrate body composition measures.

Response: We agree that defining obesity using BMI alone is a key issue. We expanded the Limitations subsection to acknowledge that using BMI as the sole criterion for obesity is a limitation, as it does not capture fat distribution or body composition. We specifically mention the lack of central obesity measures (e.g., waist circumference) and body fat percentage data. We have added a sentence suggesting that future studies should incorporate these measures to provide a more comprehensive assessment(Line 474 to Line 478).

Comment 4: Clearly report subgroup sample sizes (especially in male phenotypes) in the Results tables.

Response: We agree that clearly report subgroup sample sizes is essential. Therefore, in the revised manuscript, The sample sizes for all subgroups (with particular attention to male phenotype categories) have been explicitly specified by adding a separate row in the relevant results tables (Tables 2 and 3).

Comment 5: In the Discussion, interpret non-significant findings in males cautiously and acknowledge the possibility of limited statistical power and potential false-negative results.

Response: In the Discussion section, when interpreting the results for males, we now explicitly state that the non-significant findings should be interpreted with caution. We acknowledge that these analyses, particularly within certain phenotype strata, may have been underpowered, increasing the possibility of false-negative results(Line 431 to Line 434).

Comment 6: Expand the limitations to mention unmeasured lifestyle confounders such as diet and physical activity, and discuss their potential impact on the observed associations.

Response: We agree with the editor that we did not fully control for confounding factors.We have expanded the limitations section of the discussion as suggested. We acknowledge that unmeasured or inadequately measured potential confounders, such as dietary patterns and physical activity, may have influenced the findings presented in this paper(Line 478 to Line 482).

Comment 7: Strengthen the mechanism section by briefly discussing metabolic features that may be particularly relevant in the Chinese population, supported by existing literature.

Response: We agree it is necessary to briefly discuss the mechanisms of metabolic profile intensification that are particularly relevant to the Chinese population. We strengthened the discussion at the mechanistic level by integrating research evidence from studies conducted in Chinese and East Asian populations. We briefly explore population-specific metabolic and dietary characteristics, such as the tendency toward visceral fat accumulation at lower BMI thresholds and a predominantly high-carbohydrate diet, and how these factors differentially influence obesity tolerance phenotypes and their association with colorectal adenoma risk(Line 383 to Line 403). To support this context-specific discussion, we have supplemented relevant literature citations(Line 630 to Line 648).

Comment 8: Add a short comparison between your definition of metabolic health and other commonly used definitions.

Response: We agree with the editor for the useful suggestion.We have added a section to our discussion briefly comparing our definition with other methods (which may include waist circumference and insulin resistance). Our definition is both suitable for comparison with a large-scale of existing literature and clinically practical(Line 311 to Line 319).

Comment 9: Justify the rationale for selecting the current classification to enhance interpretability and comparability with other studies.

Response: We clearly justify our choice by stating that this definition was selected to balance clinical relevance with the available data in our study, and to enhance comparability with a significant body of existing literature that uses similar criteria, thereby facilitating the interpretation of our findings within the broader research context.(Line 311 to Line 319).

Responds to reviewer’s comments:

Comment 1: The cross-sectional design cannot establish a temporal causal relationship between metabolic obesity phenotypes and CRA. Additionally, the MHO phenotype is transient, and the existing conclusion that "MHO is not associated with CRA risk" may be limited by the duration of phenotype observation. It is recommended to further clarify in the discussion that prospective cohort studies are needed in the future to track the long-term impact of phenotype transition on CRA occurrence, avoiding conclusion bias caused by short-term observation.

Response:We fully agree with this important point. In the revised Discussion section, we have:

Clearly stated in a dedicated limitations paragraph that due to the cross-sectional design, our findings represent associations and cannot be interpreted as evidence of a temporal or causal relationship between metabolic phenotypes and CRA(Line 462 to Line 464).

Added a specific discussion on the dynamic nature of the MHO phenotype. We now explicitly note that as our study captured metabolic health at a single time point, we cannot account for potential transitions from MHO to metabolically unhealthy states over time. We have revised the text to clarify that our conclusion regarding MHO and CRA risk is necessarily constrained by this snapshot assessment. Furthermore, we strongly emphasize that future longitudinal cohort studies are essential to track these phenotype transitions and to definitively evaluate their long-term impact on CRA development, thereby avoiding bias from short-term observation. (Line 413 to Line 417).

Comment 2: Obesity is only defined by BMI, without including complementary indicators reflecting central obesity such as waist circumference and body fat percentage. However, central obesity is more closely associated with metabolic abnormalities and intestinal lesions. It is suggested to supplement the discussion on this limitation and point out that future studies should integrate body composition analysis to refine the impact of different obesity subtypes.

Response: We thank the reviewer for highlighting this key limitation. We have expanded the Limitations subsection in the Discussion to explicitly acknowledge that using BMI as the sole measure is a significant constraint. We note that BMI does not distinguish fat distribution or differentiate between fat and lean mass. We specifically mention the absence of central obesity indicators like waist circumference or body fat percentage, which are more strongly correlated with visceral adiposity, metabolic dysfunction, and, as the reviewer notes, intestinal pathology. To address this, we have added a sentence recommending that future research should integrate detailed body composition analysis to refine the understanding of how different obesity subtypes influence CRA risk(Line 474 to Line 478).

Comment 3: The sample sizes of some phenotypes in the male subgroup are relatively small, leading to insufficient statistical power which may mask true associations. It is recommended to clearly mark the sample size of each subgroup in the result tables, and emphasize in the discussion that the "no significant difference in males" should be interpreted with caution. The false negative result cannot be excluded due to limited sample size, and larger-scale studies are required to verify the biological mechanisms underlying sex differences.

Response: We appreciate the reviewer's careful attention to this detail. We have taken the following actions:

The sample size (n) for each phenotype subgroup within the male and female cohorts is now clearly presented in the separate row in the relevant results tables (Tables 2 and 3).

In the Discussion section, when interpreting the sex-specific results, we have added a clear cautionary statement. We now explicitly state that the lack of statistically significant associations observed among males should be interpreted with caution due to the relatively smaller sample sizes within some phenotype strata, which may have resulted in limited statistical power. We acknowledge that this increases the possibility of Type II error (false-negative results) and that larger-scale studies are required to confirm or refute these findings and to robustly explore the biological mechanisms underlying the apparent sex differences(Line 431 to Line 434).

Comment 4: The current confounding factors only include smoking and alcohol consumption, without incorporating key lifestyle factors such as dietary structure and physical activity, which may simultaneously affect metabolic status and CRA occurrence. It is recommended to supplement the discussion on the potential impact of this omission on the results or explain it in the limitations section.

Response: This is a valid and important point. We have added a statement to the Limitations paragraph of the Discussion. We now explicitly state that although we adjusted for several important factors, we did not have comprehensive data on other potential confounders, most notably detailed dietary habits and objective measures of physical activity. We acknowledge that unmeasured or inadequately measured potential confounders, such as dietary patterns and physical activity, may have influenced the findings presented in this paper(Line 478 to Line 482).

Comment 5: The mechanism section only generally describes the role of hyperglycemia and dyslipidemia, lacking targeted analysis of the metabolic characteristics of the Chinese population. It is suggested to briefly supplement the specificity of relevant mechanisms in the Chinese population by referencing existing literature to enhance the depth of the discussion.

Respond We thank the reviewer for this suggestion to enhance the depth of our discussion. In the revised Mechanism section of the Discussion, we have incorporated a brief, targeted discussion supported by relevant literature. We strengthened the discussion at the mechanistic level by integrating research evidence from studies conducted in Chinese and East Asian populations. We briefly explore population-specific metabolic and dietary characteristics, such as the tendency toward visceral fat accumulation at lower BMI thresholds and a predominantly high-carbohydrate diet, and how these factors differentially influence obesity tolerance phenotypes and their association with colorectal adenoma risk(Line 383 to Line 403). To support this context-specific discussion, we have supplemented relevant literature citations(Line 630 to Line 648).

Comment 6: Supplementary Explanations for Definitions and Classifications. Although the definition standards of metabolic health have been clarified, there is no universal standard for "metabolic health". It is recommended to compare the definition used in this study with other mainstream definitions in the discussion, explain the rationality of the selected standard, and enhance the comparability of results.

Respond We have addressed this point by adding a new paragraph in the discussion section. In this paragraph: We briefly compare our definition with other commonly used criteria in the literature (which may include waist circumference and insulin resistance).We justify our selection by explaining that this definition was chosen for its clinical practicality, alignment with several major epidemiological studies, and suitability for our available data. We state that this approach was taken to enhance the interpretability and comparability of our results with a substantial body of existing research(Line 311 to Line 319).

We appreciate your time and consideration, and we anticipate your favorable response at your earliest convenience.

Sincerely yours,

Dr. Qin Zhu

Professor

Department of Gastroenterology,

Zhejiang Hospital

HangZhou 310013, P. R. China

Tel: (86)-17706413537

Email: zhuqin-1@163.com

---

## [Decision Letter · Decision Letter 2]

8 Feb 2026

Association between different metabolic obesity phenotypes and colorectal adenoma

PONE-D-25-42576R2

Dear Dr. Zhu,

We’re pleased to inform you that your manuscript has been judged scientifically suitable for publication and will be formally accepted for publication once it meets all outstanding technical requirements.

Kind regards,

Muhammad Shahzad Aslam, Ph.D.,M.Phil., Pharm-D

Academic Editor

PLOS One

Additional Editor Comments (optional):

Reviewers' comments:

Reviewer's Responses to Questions

**Comments to the Author**

Reviewer #2: All comments have been addressed

2. Is the manuscript technically sound, and do the data support the conclusions?

Reviewer #2: Yes

3. Has the statistical analysis been performed appropriately and rigorously?

Reviewer #2: Yes

4. Have the authors made all data underlying the findings in their manuscript fully available?

Reviewer #2: Yes

5. Is the manuscript presented in an intelligible fashion and written in standard English?

Reviewer #2: Yes

Reviewer #2: (No Response)

**Do you want your identity to be public for this peer review?** For information about this choice, including consent withdrawal, please see our Privacy Policy

Reviewer #2: No

---

## [Editor Report · Acceptance letter]

PONE-D-25-42576R2

PLOS One

Dear Dr. Zhu,

I'm pleased to inform you that your manuscript has been deemed suitable for publication in PLOS One. Congratulations! Your manuscript is now being handed over to our production team.

Kind regards,

on behalf of

Dr. Muhammad Shahzad Aslam

Academic Editor

PLOS One